# The Role of Monoclonal Antibodies in the Treatment of Myeloma Kidney Disease

**DOI:** 10.3390/ph17081029

**Published:** 2024-08-05

**Authors:** Daniele Derudas, Sabrina Chiriu

**Affiliations:** S.C. di Ematologia e C.T.M.O. Ospedale Oncologico di Riferimento Regionale “A. Businco” ARNAS “G. Brotzu”, 09126 Cagliari, Italy; sabri.chiriu@gmail.com

**Keywords:** monoclonal antibodies, multiple myeloma, renal failure

## Abstract

Renal failure is one of the most important manifestations of multiple myeloma. It is caused by renal lesions such as cast nephropathy, immunoglobulin deposition disease, AL amyloidosis or other glomerular and/or tubular diseases, mostly due to the toxic effect of free light chains in serum. Renal failure can represent a clinical emergency and is associated with poor outcome in newly diagnosed and relapsed/refractory multiple myeloma patients. Although progression-free survival and overall survival have improved with the introduction of novel agents, renal failure remains a challenge for the treatment of patients with multiple myeloma. Monoclonal antibodies are a component of therapy for newly diagnosed and relapsed/refractory patients and, based on clinical trials and real-world experience, are also safe and effective for subjects with renal failure, even if they are on dialysis. Most of the data are on anti-CD38 and anti-SLAM7 antibodies, but new antibody–drug conjugates such as belantamab mafodotin and bispecific antibodies also appear to be effective in myeloma kidney disease. In the future, we will have to face some challenges, such as defining new criteria for renal response to treatment, defining specific trials for these difficult-to-treat patients and integrating different therapeutic options.

## 1. Introduction

Renal insufficiency (RI) is a severe complication of multiple myeloma (MM) and has been documented in up to 50% of newly diagnosed MM (NDMM) patients [1,2] and 2–4% of the relapsed/refractory (RRMM) population [3]. RI can occur in 25% of patients who do not have kidney impairment at diagnosis [4], and pre-existing renal insufficiency can also worsen during the course of the disease [3].

Kidney disease associated with MM is primarily due to the toxic effects of serum monoclonal free light chains (FLCs) on the glomeruli and tubules [5,6,7,8].

Other renal lesions are associated with the deposition or precipitation of whole monoclonal immunoglobulins or fragments thereof (e.g., AL amyloidosis, monoclonal immunoglobulin deposition, Fanconi disease). Impaired renal function can also be caused by dehydration, hypercalcemia, infections, tumor lysis syndrome and nephrotoxic drugs [9,10,11].

More recently, RI associated with MM has been defined as a serum creatinine level greater than 2 mg/dL or decreased creatinine clearance (CrCl < 40 mL/min), one or both of which are attributable to plasma cell dyscrasia [1]. Either the Modification of Diet in Renal Disease (MDRD) or Chronic Kidney Disease Epidemiology Collaboration (CKD-EPI) equations are recommended for assessing creatinine clearance by estimated glomerular filtration rate (eGFR) in this population [1,12,13]. Renal insufficiency in MM results in decreased overall survival (OS) and increased risk of early mortality [4,6,7], and several studies have demonstrated the association between clinical outcome and degree of eGFR impairment [3,6].

The introduction of novel agents, such as immunomodulators, proteosome inhibitors and monoclonal antibodies, has led to a significant improvement in both PFS, OS and kidney function compared to conventional chemotherapy. Despite these results, OS remains worse in patients with MM kidney disease compared to the population without RI at diagnosis.

The evaluation of outcomes in patients with MM renal dysfunction is complicated by several issues, such as the exclusion of this population from clinical trials and the lack of clear inclusion criteria for the degree of renal dysfunction, the lack of standardized criteria for defining RI and renal response, the difficulty in defining the true cause of renal disease, especially in the elderly, and the use of equations developed for the estimation of chronic renal disease to evaluate acute kidney injury.

Monoclonal antibodies have recently been added to the therapeutic armamentarium for NDMM and RRMM patients. Naked antibodies such as daratumumab, isatuximab and elotuzumab, in combination with other new agents, represent the gold standard for transplant-eligible and non-transplant-eligible MM patients as well as relapsed/refractory patients.

Drug-conjugated antibodies (ADCs) and bispecific antibodies have also shown promising activity in heavily treated patients.

The aim of this review is to present the available data on the efficacy and safety of monoclonal antibodies in MM patients with RI, both in clinical trials and in clinical practice. Considering the importance of the newest immunotherapies in the treatment of relapsed/refractory multiple myeloma, particular attention is given to the role of ADCs and bispecific antibodies and the issues surrounding the use of these drugs in real life for this particular populations.

## 2. Monoclonal Antibodies

### 2.1. Antibodies against CD38

Daratumumab and isatuximab are the two naked monoclonal antibodies targeting CD38, a cell surface glycoprotein that is mainly, but not exclusively, expressed on neoplastic plasma cells and is involved in the regulation of cell adhesion, intracellular calcium signaling, apoptosis, survival, proliferation and control of immune surveillance [14].Daratumumab is the first human immunoglobulin (Ig)G1 mAb to demonstrate clinical efficacy. Its role, either alone or in combination with other novel agents, has been explored in several phase 1–3 trials [15,16,17,18,19,20,21,22]. Daratumumab-based treatments are the gold standard for transplant-eligible (TE) and transplant-ineligible (TI) patients, particularly in European countries [23]. As a single agent or in combination with dexamethasone, daratumumab has shown high efficacy in patients with renal failure.

In the phase 2 DARE study, daratumumab was administered in combination with dexamethasone in a selected population of RRMM patients and patients with severe RI (eGFR < 30 mL/min per 1–73 m^2^ or on dialysis). The study included 38 patients with an eGFR < 30 mL/min per 1–73 m^2^ with an overall response rate (ORR) of 47% and a 6-month progression-free survival (PFS) of 54%. The 17 dialysis-dependent patients showed an ORR of 47%, and the renal response rate was 18% in patients with an eGFR < 30 mL/min per 1–73 m^2^ [24].The efficacy of daratumumab monotherapy in terms of ORR was demonstrated in a pooled analysis of the pivotal phase 1/2 trial and the supportive phase 2 trial in RRMM patients with a CrCl of 30–60 mL/min [15,25]. In addition, several case reports [26,27,28] and case series [29,30] of daratumumab-based associations showed clinical efficacy and reduced dialysis frequency or dialysis independence in dialysis-dependent RRMM patients.

Daratumumab-based treatments demonstrated efficacy and safety in NDMM patients.

c.In the phase 3 CASSIOPEIA [20] study for TE NDMM patients, the group with the quadruple therapy daratumumab–bortezomib–thalidomide–dexamethasone (Dara-VTD) improved in terms of ORR and PFS compared to the triple VTD without the monoclonal antibody in TE patients. The cut-off value for creatinine clearance was 40–90 mL/min. Another phase 2 study investigated the combination of daratumumab–bortezomib–lenalidomide–dexamethasone in NDMM patients with severe renal impairment (CrCl < 30 mL/min) with regard to efficacy and safety [31].Of thirteen patients included, seven achieved a CrCl of ≥ 50 mL/min after two cycles (median C3D1 CrCl 61 mL/min) and all had an improvement in serum creatinine at C3D1. The overall response rate was 100% and the ≥ VGPR rate was 82%. The best responses were complete response (three), very good partial response (six) and partial response (two).d.Recently, another real-word report [32] with this combination in a population with kidney impairment showed encouraging results. Among 18 patients evaluated for response, the ORR was 94.4% and the median duration of response was 2 months. After induction therapy, the ORR was 100% and 50% of patients had a complete response. The data reported after consolidation showed that all patients achieved a very good partial response (VGPR) and a complete response (CR) with minimal residual disease negativity by next-generation sequencing (NGS MRD). Fifty-five patients (10/18) achieved renal remission after one cycle, 86.7% after two cycles, 91.7% after three cycles and 100% after four cycles.

The combination of daratumumab with lenalidomide and dexamethasone (Dara-Rd) and daratumumab with bortezomib–melphalan–prednisone (Dara-VMP) represent two of the therapy option for TI NDMM patients.

e.The phase 3 ALCYONE [33] study showed that the association Dara-VMP was able to improve overall response rates, MRD negativity rates and PFS without safety issues in patients with multiple myeloma and creatinine clearance of 40–60 mL/min compared to VMP alone.f.A trial with Dara-Rd (MAIA) [34] was also associated with improved survival outcomes compared to lenalidomide–dexamethasone in newly diagnosed multiple myeloma patients with a CrCl of 30–60 mL/min. Various daratumumab-based treatments are available for RRMM patients.

A different combination could be used for RRMM with renal disease.

g.The CASTOR [34] study showed a higher efficacy in terms of PFS for the combination of daratumumab–bortezomib–dexamethasone compared to a doublet bortezomib–dexamethasone (VD) in a population with a creatinine clearance of 20–60 mL/min.h.A further association of daratumumab with carfilzomib and dexamethasone (Dara-Kd) (phase 3 study CANDOR [35]) demonstrated prolonged PFS in a cohort of patients with a CrCl of 15–50 mL/min. A subgroup analysis of PFS by baseline level of renal function was conducted (≥15 to 50, ≥50 to < 80 and ≥80 mL/min). Accordingly, only 12% of patients (38 of 311) in the Dara-Kd group and 18% (27 of 154) in the carfilzomib–desametasone (Kd) group had a CrCl ≥ 15 to <50 mL/min.i.The Dara-Rd (phase 3 POLLUX trial [36]) association increased PFS compared to lenalidomide–dexamethasone in patients with a CrCl of 30–60 mL/minj.Finally, in the phase 3 APOLLO [37] study, the combination of daratumumab with pomalidomide and dexamethasone also showed superiority in terms of PFS compared to the duplet pomalidomide and low-dose dexamethasone in patients with RI (creatinine clearance of 30–60 mL/min).

Isatuximab is a new-generation monoclonal antibody against the CD38 receptor. Compared to daratumumab, this agent shows a particular ability to induce direct apoptosis of malignant cells [38].

k.The combination of isatuximab–pomalidomide–dexamethasone (Isa-Pd) was used in the ICARIA-MM [39] study in comparison with the dual combination of pomalidomide and low-dose dexamethasone (Pd) in RRMM patients. The inclusion criterion was an eGFR ≥ 30 mL/min/1.73 m^2^. A subgroup analysis of patients with RI (eGFR <60 mL/min/1.73 m^2^) [40] was performed to assess efficacy and safety in this population: of 287 patients with evaluable eGFR at baseline, only 55 (38.7%) in the Isa-Pd group and 49 (33.8%) in the Pd group had RI. A PFS benefit of Isa-Pd and a higher ORR and MRD negativity rate in patients with RI compared to Pd was described. This subgroup analysis, unique among phase 3 anti-CD38 monoclonal antibody trials, examined renal response rates and rates of adverse events in patients with RI at baseline. The rates of complete response to renal treatment were 71.9% with Isa-Pd and 38.1% with Pd. Isa-Pd also resulted in improved time to renal response. In this population with RI, grade ≥ 3 adverse events and treatment-emergent adverse events (TEAEs) occurred more frequently in the Isa-Pd group. However, after adjusting for the increased treatment exposure in the Isa-Pd group, the number of serious TEAEs per patient-year in patients with RI was similar in both groups [40].l.Recently, isatuximab was also approved in combination with carfilzomib and dexamethasone (Isa-Kd) for the treatment of patients with RRMM (IKEMA trial 41). The limit for inclusion in the study was the recruitment of patients with an eGFR of only 15 mL/min/1.73 m^2^ [41]. In a pre-specified subgroup analysis [42], efficacy, renal response and safety were assessed in patients with RI (defined as eGFR < 60 mL/min/1.73 m^2^) at the time of interim analysis. Patients with RI (43 in the Isa-Kd arm and 18 in the Kd arm) accounted for 26.1% and 16.2% of patients with evaluable eGFR at baseline, respectively. Approximately 2.5% of patients in each arm had an eGFR of ≥15 to <30 mL/min/1.73 m^2^. In these patients, an improvement in PFS benefit and higher complete response and MRD negativity rates were observed in the Isa-Kd arm. Complete renal response rates and time to first renal response improved with Isa-Kd compared to Kd. An eGFR < 30 mL/min/1.73 m^2^ at baseline appears to have a negative impact on the ability to achieve a good renal response with isatuximab. Isa-Kd showed a manageable safety profile in patients with and without RI. The presence of RI was not associated with a higher rate of grade 3 or higher heart failure due to the effect of carfilzomib 42.m.In real life, different experiences with isatuximab in patients with RI have been described. A case report [43] of a patient on dialysis described an RRMM patient who was treated with Isa-Pd after seven prior lines of therapy. The free light chain λ level dropped from 2070 mg/L to 412 mg/L 12 d after starting treatment with Isa-Pd. No infusion reactions or clinically significant decreases in white blood cell counts were documented during treatment. The duration of response was seven cycles.

The results of the main studies with daratumumab and isatuximab are summarized in Table 1.

### 2.2. CS1 Antibodies

Elotuzumab is a humanized monoclonal antibody directed against the signaling lymphocyte activation molecule-7 (SLAMF7) or CS-1. This antibody is currently available for the treatment of RRMM patients in combination with Rd and Pd, according to the results of the randomized phase 3 ELOQUENT-2 [45] and ELOQUENT-3 [46] trials, respectively. 

In ELOQUENT-2 and in ELOQUENT-33, respectively, patients with a CrCl ≥ 30 mL/min and ≥45 mL/min were eligible, but neither trial reported safety or efficacy data stratified by renal function.A small phase 1b study [46], showed a good tolerability and efficacy of elotuzumab for the treatment of MM patients with RI, including end-stage renal disease. Three levels of renal function were assessed: normal (CrCl ≥ 90 mL/min, in eight patients), severely impaired (CrCl < 30 mL/min, not requiring dialysis, in nine patients) and end-stage (requiring dialysis, in nine subjects). Overall, 75%, 67% and 56% of patients in the three groups responded to treatment. Two patients in the severely impaired group (including one with RRMM) showed a mild renal response. Notably, no difference was observed between the groups in terms of grade 3/4 adverse events.

### 2.3. Antibody–Drug Conjugates (ADCs)

Belantamab mafodotin is the first drug-conjugated antibody (ADC) used for MM treatment and it targets B-cell maturation antigen (BCMA)-expressing myeloma cells [47]. Belantamab mafodotin received FDA and EMA approval for patients with RRMM based on the phase 2 DREAMM-2 [48] study, which established an eGFR ≥ 30 mL/min/1.73 m^2^ at screening for enrollment. A post hoc analysis was conducted to evaluate efficacy and survival as a function of renal function (normal as eGFR ≥ 90 mL/min/1.73 m^2^, mildly impaired as ≥60 to <90 mL/min/1.73 m^2^ and moderately impaired as ≥30 to <60 mL/min/1.73 m^2^) [49]. ORR and median PFS were similar in the different renal function groups. No difference in the rates of keratopathy and grade 3/4 adverse events was observed between the different groups. No data, in order of efficacy in this setting of RRMM, are available for the combination with proteosome inhibitors and immunomodulant agents recently published.

### 2.4. Bispecific Antibodies

Bispecific antibodies are novel immunotherapeutics that have been shown to be effective in heavily treated patients with multiple myeloma. They form immune synapses between the T-cell surface marker CD3 and malignant plasma cell markers such as the B-cell maturation antigen (BCMA), FcRH5 and the G-protein-coupled receptor (GPRC5D).

The first compounds to be used in clinical trials are teclistamab and elnartamab, which target BCMA, and talquetamab, which targets GPRC5D. The pharmacokinetics of teclistamab were not significantly affected by mild or moderate renal impairment, as stated in the prescribing information [50]. Based on the general principles of excretion and clearance of monoclonal antibodies and the pharmacokinetics documented for teclistamab, patients with severe renal impairment, including ESRD patients on hemodialysis, should be able to receive this agent without significant change in pharmacokinetics [51].

The MajesTEC-1 study, which evaluated the safety and efficacy of this bispecific antibody as a monotherapy for RRMM subjects, excluded patients with an estimated glomerular filtration rate <40 mL/min, and 73% of patients had an eGFR > 60 mL/min [52]. Recently, two real-world experiences on the efficacy and safety of teclistamab in patients with RRMM and renal failure have been published.The first real-world experience of treatment with teclistamab was reported for seven patients with severe renal failure, including four patients with end-stage renal disease (ESRD) who were on dialysis at the time of initiation of therapy [53]. The requirements of the Risk Evaluation and Mitigation Strategy (REMS) were met. All patients were hospitalized for doses within the step-up phase. No dose modifications were made for cycles 1 and 2, and for patients undergoing hemodialysis, the dose was administered after the procedure. In the outpatient setting, the target doses were administered on a day when dialysis was not performed. In cycle 3, the dosing frequency was changed to every 2 weeks for eight doses, then every 4 weeks from cycle 7 onwards. At the beginning of teclistamab therapy, infection prophylaxis with oral levofloxacin and treatment against herpes simplex viruses, varicella zoster viruses and Pneumocystis jirovecii was started. From the second month onwards, a monthly infusion of intravenous immunoglobulin (IVIG) was scheduled. The median follow-up time was 2 months (range 1–7). Of the population, five showed a response without signs of increased toxicity and continued treatment with signs of a very good partial response (VGPR). Two patients who experienced disease progression discontinued treatment before the second cycle. Of note, both patients had extramedullary disease. In terms of safety, the incidence and grading of CRS in these patients were consistent with those described for teclistamab. Neither immune effector cell-associated neurotoxicity syndrome (ICANS) nor infections were reported in these patients. In the only patient with acute renal failure related to myeloma progression, there was a significant improvement in renal function (eGFR from 27 mL/min to 56 mL/min after one cycle). In one patient, renal function was stable; in another, renal function deteriorated with disease progression.In the second report [54], thirteen patients with ESRD requiring hemodialysis were treated with teclistamab as a monotherapy in French hospitals. The median time since diagnosis was 6 years (range 2–9) and the median number of previous lines of therapy was 4 (range 3–9). Most patients had myeloma-related ESRD. Of the patients, 75% had light chain multiple myeloma. All evaluable patients achieved a response and achieved a VGPR or better. No progression or deaths were reported at a median follow-up of 4 months. The step-up dosing regimen was the same as that used in a population without renal failure, but the drug was administered within 1–2 d of dialysis. Half of the tocilizumab-treated patients developed grade 1 or 2 CRS, with only one patient receiving additional dexamethasone. No ICANS was reported. The infection rate does not appear to be higher than in non-dialysis patients with grade <3 events, with the exception of one patient who developed a severe COVID-19 infection and Pseudomonas aeruginosa septicemia, which resolved after discontinuation of teclistamab and specific treatments. Considering the small groups and short follow-up period, these reports have shown that teclistamab is a viable, effective and safe option for patients with RRMM with severe renal impairment and ESRD requiring dialysis.

No formal studies have been conducted for elnartamab and talquetamab in patients with renal impairment. Pharmacokinetic analyses indicate that mild (eGFR 60–90 mL/min) or moderate (eGFR 30–60 mL/min) renal impairment does not significantly affect the pharmacokinetics of both antibodies. No data are available for patients with severe renal impairment (eGFR < 30–60 mL/min) [55,56]. A case report was recently published on the efficacy and safety of elnartamab in an RRMM patient with ESRD on hemodialysis [57].

## 3. Discussion

Renal dysfunction is a common complication of MM, either at the onset of the disease or in the relapsed/refractory stage, and has a negative impact on survival. Treatment options for patients with MM are rapidly increasing and overall survival is improving, but to date, the therapeutic strategy and best association for patients with myeloma kidney disease is unknown. Monoclonal antibodies are critical for the treatment of MM populations at any point in the disease course, whether as a monotherapy or in combination with other antineoplastic agents. In particular, naked monoclonal antibodies against CD38 (daratumumab and isatuximab) and SLAMF7 (elotuzumab) are currently being integrated into the treatment of TE and TI NDMM patients as well as RRMM populations. The agents in combination have resulted in a profound and durable response with improved patient survival. In the population with RI, daratumumab showed efficacy in terms of hematologic and renal response. Its role has been confirmed in clinical trials and in practice in NDMM and RRMM patients. Transplant eligibility has no impact on efficacy and safety. None of these naked monoclonal antibodies require dose adjustment according to eGFR, even in dialysis-dependent ESRD patients, unlike the agents used in combination, and renal failure should not represent a contraindication in the treatment of this population of patients (see Table 2).

The choice of the best combination must be made according with the therapy history and patient and disease characteristics.

The most recent IMWG guidelines for the treatment of renal dysfunction associated with multiple myeloma recommend the use of daratumumab, isatuximab-based therapy and elotuzumab in combination with lenalidomide in patients with moderate to severe renal insufficiency.

Regarding the latest generation of monoclonal antibodies such as ADCs (belantamab mafodotin) and bispecific antibodies (teclistamab, elnartamab and talquetamab), the pivotal clinical trial and prescribing information have shown intact efficacy and acceptable toxicity in heavily treated RRMM patients with mild or moderate renal insufficiency. Few clinical cases have reported data on the administration of these treatments in dialysis-dependent renal insufficiency.

Despite these data, several aspects must be taken into account when evaluating the results of the various studies with monoclonal antibodies in this population.

First of all, there is no uniform definition of RI between studies.

a.There are different formulas for calculating the estimated glomerular filtration rate (eGFR; mL/min/1.73 m^2^) [1]. In addition, inappropriate equations developed for estimating renal function in CKD may be used in patients with acute kidney injury.b.Most of these studies are not designed to detect differences between treatment arms for patients with RI.c.Patient inclusion cut-offs may vary from study to study: for this reason, many studies exclude patients with moderate renal impairment or worse. In this case, the results of the study may not accurately reflect the efficacy of the monoclonal antibody association in the RI population. Translating these data into practice may be challenging.d.Few studies have provided data on response to renal disease, and response to renal disease is also not standardized according to IMWG criteria. Where possible, renal response should be considered in the design of clinical trials and subgroup analyses should be published to determine the best treatment for patients with RI. The IMWG defined different degrees of renal response:Complete renal response (CrR as an increase in baseline eGFR to ≥60 mL/min).Partial renal response as an increase in eGFR from a baseline of <15 mL/min to 30–59 mL/min.Minor renal response as an increase from <15 mL/min to 15–20 mL/min or, if baseline eGFR is 15–29 mL/min, to 30–59 mL/min [58].

According to the definition of minor renal response, it is not possible to achieve a partial response by definition. Various efforts are being made to revise the criteria for renal response.

e.An imprecise differential diagnosis of renal impairment in patients with multiple myeloma occurs in many cases [59].f.Therefore, between 2005 and 2019, there has been no improvement in the reported inclusion criteria related to renal dysfunction, the prevalence of renal failure in the included population and outcomes in patients with RI.

## 4. Conclusions

To date, naked monoclonal antibodies, ADCs and bispecific antibodies form the backbone of therapy for MM in every setting, from the newly diagnosed TE and TI populations to RRMM patients. Data from subgroup analyses of clinical trials and real-world experience have shown that monoclonal antibodies are efficacious in patients with RI, including patients with dialysis-dependent ESRD; they improve PFS compared to previous treatments and are well tolerated without the need for dose adjustment.

## 5. Future Directions

To improve knowledge of the efficacy and safety of monoclonal antibodies in the treatment of MM with renal insufficiency in the future, it is essential to include renal response in the design of clinical trials, and it is essential to design specific studies to prospectively evaluate outcomes in large groups of patients with renal insufficiency (including patients requiring dialysis).

## Figures and Tables

**Table 1 pharmaceuticals-17-01029-t001:** Results of studies with daratumumab and isatuximab.

	Cut-Off Renal Impairment	Median Progression-Free Survival		Median Overall Survival		Overall Response Rate (%)	Complete Renal Response (%)	Grade ≥ 3 Adverse Events
		Months	HR (95%CI)	Months	HR (95%CI)			
ALCYONE DaraVMp VMp	≥30 to <60	NR 16.9	0.36 (0.24–0.56)	NR NA	NA NA	89 73	NA NA	47 42
CASSIOPEIA DaraVtd Vtd	≥40 to <90	NA NA	0.37 (0.21–0.66)	NA NA	NA NA	NA NA	NA NA	NA NA
MAIA (len 25 mg) DaraRd Rd	≥30 to <60	NR 35.4	0.42 (0.24–0.72)	NR NR	0.37 (0.21–0.66)	NA NA	NA NA	NA NA
MAIA (len < 25 mg) DaraRd Rd	≥30 to <60	49.1 24.9	0.56 (0.38–0.83)	62.8 54.8	0.81 (0.52–1.26)	NA NA	NA NA	NA NA
POLLUX DaraRd Rd	≥30 to <60	33.6 11.3	0.41 (0.26–0.65)	NR NR	NA NA	91 68	NA NA	NA NA
CASTOR DaraVd Vd	≥20 to <60	NR 6.5	0.55 (0.30–1.02)	NA NA	NA NA	NA NA	NA NA	NA NA
APOLLO DaraPd Pd	≥30 to <60	12.1 6.1	0.59 (0.35–0.99)	NA NA	NA NA	NA NA	NA NA	NA NA

Adapted from [44].

**Table 2 pharmaceuticals-17-01029-t002:** Dose modification of monoclonal antibodies and agents in combination.

Creatinine Clearance					On Dialysis
	≥60 mL/min	30–59 mL/min	15–29 mL/min	<15 mL/min	
Dexamethasone	20–40 mg	No dose modification	No dose modification	No dose modification	No dose modification
Melphalan	0.15–0.25 mg/kg per day orally; 200 mg/m^2^ intravenously	Reduction by 25% orally; high dose: 140 mg/m^2^ intravenously	Reduction by 25% orally; high dose: 140 mg/m^2^ intravenously	Reduction by 50% orally; high dose: 140 mg/m^2^ intravenously	Reduction by 50% orally; high dose: 140 mg/m^2^ intravenously
Bortezomib	1.3 mg/m^2^	No dose modification	No dose modification	No dose modification	No dose modification
Carfilzomib	Full dose: 20/27 mg/m^2^ 20/56 mg/m^2^ 20/70 mg/m^2^	No dose modification	No dose modification	No dose modification	No dose modification, after dialysis
Ixazomib	4 mg	4 mg	3 mg	3 mg	3 mg
Thalidomide	50–200 mg	No dose modification	No dose modification	No dose modification	No dose modification
Lenalidomide	25 mg per day	10 mg per day, can be increased to 25 mg per day if no toxicity occurs	15 mg every other day or 10 mg per day, can be increased to 15 mg per day if no toxicity occurs	5 mg per day, can be increased to 15 mg per day if no toxicity occurs	5 mg per day after dialysis, can be increased to 15 mg per day if no toxicity occurs
Pomalidomide	4 mg	No dose modification	No dose modification	No dose modification	No dose modification
Daratumumab	16 mg/kg intravenously or 1800 mg subcutaneously	No dose modification	No dose modification	No dose modification	No dose modification
Isatuximab	10 mg/kg	No dose modification	No dose modification	No dose modification	No dose modification
Elotuzumab	10 mg/kg	No dose modification	No dose modification	No dose modification	No dose modification
Belantamab mafodotin	2.5 mg/kg	No dose modification	No dose modification	No dose modification	No dose modification
Teclistamab	1.5 mg/kg	No dose modification with creatinine clearance > 40 mL/min	Not determined yet	No dose modification	Dose modification not determined yet

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
