# Peer review of "The Role of Monoclonal Antibodies in the Treatment of Myeloma Kidney Disease"

_pharmaceuticals, 2024, doi:10.3390/ph17081029_

Round 1
Reviewer 1 Report
Comments and Suggestions for Authors
This manuscript is about the role of “The role of monoclonal antibodies in the treatment of myeloma
kidney disease”. This can help to learn about the mechanisms by how monoclonal antibodies play their role. The manuscript can extend our knowledge in the field and provide an overview of current research about the biological effects of monoclonal antibodies on myeloma kidney disease.
Overall, I thoroughly enjoyed reviewing it, but I have several requests for revision.
The authors address the following comments:
-Please change keywords based on PubMed Mesh terms and rearrange them according to the English alphabet.
-The article is written very well and carefully. However, some punctuation and grammatical problems in the text can be modified.
-It is necessary for the authors to clarify the aims of this study more.
-It is suggested to mention the biological roles of monoclonal antibodies in some diseases.
-It was detected as high plagiarism, which needs to be corrected.
The article is written very well and carefully. However, some punctuation and grammatical problems in the text can be modified.D
Author Response
Comment 1- Please change keywords based on PubMed Mesh terms and rearrange them according to the English alphabet.
Reply 1 - Done-The article is written very well and carefully. However, some punctuation and grammatical problems in the text can be modified.
Comment 2: It is necessary for the authors to clarify the aims of this study more.
Reply 2 _ Done at the bottom of introduction
Comment 3: It is suggested to mention the biological roles of monoclonal antibodies in some diseases.
Reply 3: Thanks for the comment. I decided to not go into detail of the biological role of MoAbs because I thought it was off-topic
Comment 4: It was detected as high plagiarism, which needs to be corrected.
Reply 4: Thanks. It is veriìy difficult to not cite IWMG guidelines. I tried to minimize this problem
Reviewer 2 Report
Comments and Suggestions for Authors
In this investigation by chiriu Sabrina and Derudas Daniele the role of monoclonal antibodies in the treatment of Myeloma kidney disease is described. Although the review is very specialized I recommend publication after the following minor revisions have been applied.
Please define renal insufficiency as RI (first line).
I suppose every 2 of m2 should be superscript
and a creatinine clearance of 40–90 ml/min…(l must be a capital letter)
only 15 mL/min/1.73 m241. (space after m2)
Caption of Table 2 must be on next page
Use the good template for references
Comments on the Quality of English LanguageModerate editing of the English language required
Author Response
Comment 1: Please define renal insufficiency as RI (first line).
Reply 1: Done
Comment 2: I suppose every 2 of m2 should be superscript
Reply 2: Done
Comment 3: and a creatinine clearance of 40–90 ml/min…(l must be a capital letter)
only 15 mL/min/1.73 m241. (space after m2)
Reply 3: Done
Comment 4: Caption of Table 2 must be on next page
Reply 4: Done
Comment 5: Use the good template for references
Reply 5: I followed the rules for reference. But I will double-check, thanks
Reviewer 3 Report
Comments and Suggestions for Authors
The authors investigate renal failure in survival context. Monoclonal antibodies, including anti-CD38 and anti-SLAM7 antibodies, are effective and safe for patients with renal failure, even those on dialysis. the paper states that emerging therapies like belantamab mafodotin and bispecific antibodies show promise. This work may add to the overall literature, however, I have the following comments:
-can different means (p-values ) be added to the tables?
-what are the limitations of this study?
- I suggest doing survival analysis using Kaplan-Meier or cox-regression models.
Comments on the Quality of English Language
English minor editing is required
Author Response
Comment 1: can different means (p-values ) be added to the tables?
Reply 1: I do not have add he means to not make the tables difficult to read
Comment 2: what are the limitations of this study?
Reply 2: Thank you for the comment. I have described the limitations of the trials and studies more than the review.
Comment 3: I suggest doing survival analysis using Kaplan-Meier or cox-regression models.
Reply 3: Done with HR
Reviewer 4 Report
Comments and Suggestions for Authors
1. The reference citation number in the main text cause confusion for readers. Please either make the number superscript or put the number in a bracket.
2. Is Table 1 completed? The table format is not presented well in the main text. Please revise it.
3. In the section “2.3 The format is not presented well in the main text”, looks like there are two spaces between “used” and “for” in the first sentence. Please double check it.
4. The discussion section needs to be more conclusive to discuss the monoclonal antibodies for rental failure. Please try to group related information together for better logical flow.
5. Can authors add one paragraph to compare difference of antibodies and address their advantages comparing to others?
Comments on the Quality of English LanguageEnglish needs minor edition to make it precise and to the point.
Author Response
Comment 1: The reference citation number in the main text cause confusion for readers. Please either make the number superscript or put the number in a bracket.
Reply 1: Done
Comment 2: Is Table 1 completed? The table format is not presented well in the main text. Please revise it.
Reply 2: Done
Comment 3: In the section “2.3 The format is not presented well in the main text”, looks like there are two spaces between “used” and “for” in the first sentence. Please double check it.
Reply 3: Done, thank you.
Comment 4: The discussion section needs to be more conclusive to discuss the monoclonal antibodies for rental failure. Please try to group related information together for better logical flow.
Reply 4: I tried to summarize first the efficacy and then the limitations. But I will check.
Comment 5: Can authors add one paragraph to compare difference of antibodies and address their advantages comparing to others?
Ready 5: Thank fo the suggestion. I added two phrases about the lack of superiority